# KOI: Accelerating Online Imitation Learning via Hybrid Key-state Guidance

**Jingxian Lu**[1,*], **Wenke Xia**[1,2,*†], **Dong Wang**[2,‡], **Zhigang Wang**[2], **Bin Zhao**[2,3], **Di Hu**[1,‡], **Xuelong Li**[2,4]

[1]Gaoling School of Artificial Intelligence, Renmin University of China
[2]Shanghai Artificial Intelligence Laboratory  [3]Northwestern Polytechnical University
[4]Institute of Artificial Intelligence, China Telecom Corp Ltd

**Abstract:** Online Imitation Learning struggles with the gap between extensive online exploration space and limited expert trajectories, hindering efficient exploration due to inaccurate reward estimation. Inspired by the findings from cognitive neuroscience, we hypothesize that an agent could estimate precise task-aware reward for efficient online exploration, through decomposing the target task into the objectives of "what to do" and the mechanisms of "how to do". In this work, we introduce the hybrid Key-state guided Online Imitation (KOI) learning method, which leverages the integration of semantic and motion key states as guidance for reward estimation. Initially, we utilize visual-language models to extract semantic key states from expert trajectory, indicating the objectives of "what to do". Within the intervals between semantic key states, optical flow is employed to capture motion key states to understand the mechanisms of "how to do". By integrating a thorough grasp of hybrid key states, we refine the trajectory-matching reward computation, accelerating online imitation learning with task-aware exploration. We evaluate not only the success rate of the tasks in the Meta-World and LIBERO environments, but also the trend of variance during online imitation learning, proving that our method is more sample efficient. We also conduct real-world robotic manipulation experiments to validate the efficacy of our method, demonstrating the practical applicability of our KOI method. Videos and code are available at https://gewu-lab.github.io/Keystate_Online_Imitation/.

**Keywords:** Online Imitation Learning, Robotics Manipulation

## 1 Introduction

Imitation Learning has achieved significant success in various robotic manipulation tasks through offline and online modes [1, 2, 3, 4]. The offline approach [5, 6, 7, 8, 9] seeks to learn policy by mimicking state-action pairs from expert demonstrations through supervised learning. Due to its independence from interaction with environments, offline imitation learning necessitates extensive robotic data, which requires expensive costs associated with real-world data acquisition [10, 11, 12]. In contrast, online imitation learning [13, 14], driven by a reward function informed by expert demonstrations, is able to formulate policies via exploration. Although it has the potential to refine policies with minimal expert input, the considerable gap between substantial online exploration space and limited expert trajectories challenges the estimation of exploration reward, which hinders the efficiency of online imitation learning. To estimate fine-grained reward, Optimal Transport (OT) [15, 16, 17, 18] is proposed to facilitate trajectory-matching reward by measuring the distance between exploration trajectories and expert demonstrations. However, their insufficient grasp of the semantic comprehension of tasks makes them easily converge on sub-optimal solutions [15].

---

*Equal contribution †Work is done during internship at Shanghai Artificial Intelligence Laboratory ‡Corresponding Author (wangdong@pjlab.org.cn, dihu@ruc.edu.cn)

8th Conference on Robot Learning (CoRL 2024), Munich, Germany.

Pivotal research in cognitive neuroscience [19, 20] suggests that humans could enhance cognitive processing and learning efficiency via task decomposition. Inspired by these findings, we hypothesize that agents could optimize online imitation learning efficiency by task decomposition. Specifically, by gaining a holistic grasp of both the objectives ("what to do") and the mechanisms ("how to do") of the task, an agent could be encouraged to conduct task-aware exploration.

To this end, we propose the hybrid Key-state guided Online Imitation (KOI) learning method, which effectively extracts the semantic and motion key states from expert trajectory to refine the reward estimation. We initially utilize the rich world knowledge of visual-language models to extract semantic key states from expert trajectory, clarifying the objectives of "what to do". Within intervals between semantic key states, optical flow is employed to identify essential motion key states to comprehend the dynamic transition to the subsequent semantic key state, indicating "how to do" the target task. By integrating both types of key states, we adjust the importance weight of expert trajectory states in OT-based reward estimation to empower efficient online imitation learning.

To validate the efficiency of our KOI method, we conducted comprehensive experiments across 6 Meta-World manipulation tasks and 3 long-horizon LIBERO tasks. The analysis of experiments highlights the strength of our semantic and motion key state guidance, which enhances the sample efficiency of KOI compared to previous methods. In the end, real-world experiments are conducted to prove that our method could also generalize to real complex scenarios.

## 2 Related Work

**Online Imitation Learning.** Online Imitation Learning seeks to develop policies through online exploration, guided by a reward function informed by expert demonstrations [21, 22, 23, 24]. Recognizing the difficulty of obtaining state information in real-world settings, some researchers [25, 26] explored the use of visual and language cues to inform task-aware reward estimation. However, these works mainly focused on task semantic knowledge, overlooking detailed trajectory information. In this study, we adopt the trajectory-matching reward estimation method, integrating task semantic comprehension with fine-grained trajectory motion for efficient online exploration.

**Optimal Transport for Imitation Learning.** Optimal Transport (OT) is a computational framework that addresses the problem of finding the most efficient way to transform one distribution into another, minimizing a predefined cost for this transformation [27, 28, 29]. Prior works borrow various distance metrics to measure the distance between the online exploration trajectories and expert demonstration for fine-grained imitation reward estimation [17, 18, 30, 15]. However, they regrettably ignore the semantic information of manipulation tasks, leading to convergence on suboptimal solutions [15]. In this work, we propose to combine the fine-grained trajectory-matching reward with semantic and motion key states, facilitating the efficiency of online imitation learning.

## 3 Method

During online imitation learning, a significant challenge lies in the gap between extensive online exploration space and limited expert trajectories, resulting in inefficient exploration due to imprecise imitation reward estimations. In this work, to precisely estimate task-aware reward, we extract the hybrid key states from semantic and motion aspects by the Semantic Decomposition Module (SDM) and Motion Capture Module (MCM), indicating "what to do" and "how to do" for the target task. By integrating both types of key states, we adjust the importance weights of expert trajectory within the OT-based reward estimation method, enhancing efficiency of online imitation learning.

### 3.1 Background

The goal of online imitation learning is to train a policy $\pi^e$ through exploration with the reward $r^e$ informed by the expert demonstrations $T^d$. The $i$-th state from exploration trajectory $T^e$ is denoted as $s_i^e$, and the $j$-th state from expert demonstration $T^d$ is denoted as $s_j^d$. The similarity between them

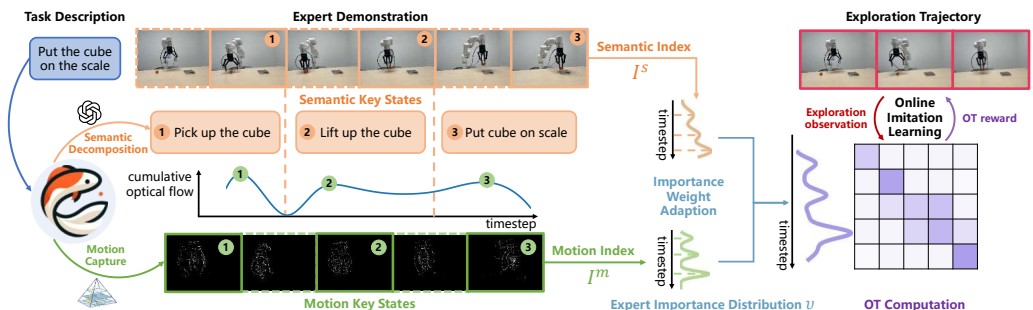

Figure 1: The pipeline of our hybrid Key-state guided Online Imitation (KOI) learning method. We first extract semantic key states with the Semantic Decomposition Module. Within intervals between semantic key states, the Motion Capture Module is proposed to identify motion key states. Further, we adjust the importance weight in OT-based reward estimation with these hybrid key states, to enable task-aware exploration for efficient online imitation learning. The color intensity of OT matrix represents the value of estimated reward.

is measured by function $\mathcal{C}$, which is defined as cosine similarity in this paper. Accordingly, the cost of transporting from source $s_i^e$ to destination $s_j^d$ is calculated as follows:

$$c_{ij} = 1 - \mathcal{C}(s_i^e, s_j^d). \tag{1}$$

Given the cost above, methods based on Optimal Transport (OT) [15, 16] aim to find the optimal transport matrix $\eta^*$ to estimate reward $r^e$ of each state. For example, the reward of exploration state at timestep $i$ is estimated using the subsequent formula:

$$r^e(s_i^e) = -\sum_{j=1}^{N^d} \eta_{ij}^* c_{ij}, \tag{2}$$

where $\eta \in \mathbb{N}^{N^e \times N^d}$ and each element $\eta_{ij}$ represents the amount of data transported from $i$ to $j$. $N^e$ and $N^d$ are lengths of exploration trajectory $T^e$ and expert demonstration $T^d$, respectively. The optimal transport matrix $\eta^*$ is obtained by measuring the distance between importance distribution $\mu$ of exploration trajectory $T^e$ and importance distribution $\nu$ of expert demonstration $T^d$, with Sinkhorn algorithm [27] employing an entropy regularization $H$ to enhance computational efficiency:

$$\eta^* = \min_{\eta} \left\{ \sum_{i=1}^{N^e} \sum_{j=1}^{N^d} \eta_{ij} c_{ij} - \frac{1}{\lambda} H(\eta) \right\}$$

$$\text{s.t.} \quad \sum_{j=1}^{N^d} \eta_{ij} = \mu_i \quad \text{for } i = 1, \dots, N^e, \tag{3}$$

$$\sum_{i=1}^{N^e} \eta_{ij} = \nu_j \quad \text{for } j = 1, \dots, N^d.$$

However, previous OT-based methods assume the distribution of expert demonstration as a uniform distribution for each state, i.e., $\nu = \frac{1}{N^d}\mathbf{1}$. While this assumption facilitates fine-grained rewards, it overlooks the importance among states. Such oversight hampers exploration efficiency for lack of task-aware state comprehension, as revealed in the qualitative analysis of Section 4.4.

To integrate trajectory-matching reward with task-aware information, we extract the semantic key states, highlighting the objectives of "what to do", and identify the motion key states, revealing the mechanisms of "how to do'. With the guidance of both types of key states, we adjust the importance distribution of expert trajectory $\nu$ for accurate reward estimation as described in Section 3.4, which encourages the agent to conduct task-aware exploration to facilitate online imitation learning.

## 3.2 Semantic Key-state Extraction

Foundation Models have demonstrated their rich world knowledge and strong generalization capabilities in various visual-language tasks [31, 32]. In this work, we propose utilizing GPT-4v to design the Semantic Decomposition Module (SDM) for task-specific subgoal extraction.

Concretely, we initially instruct GPT-4 to decompose the task description $L^t$ into $K$ subgoals, denoted as $L^s = \{l_1^s, l_2^s, .., l_K^s\}$. Subsequently, GPT-4v is employed to locate corresponding semantic key states of the expert demonstration. To manage the number of input images, we sample states every $T$ step from expert observations, forming the query sets $O$. To ensure the temporal consistency of located key states, both sampled expert observation $O$ and subgoal language descriptions $L^s$ are fed into GPT-4v to get the semantic key state index set $I^s$:

$$I^s = GPT\text{-}4v(O, L^s). \tag{4}$$

Consequently, SDM produces a temporally coherent sequence of indices $I^s$ that precisely match the semantic key states to subgoal descriptions, clearly defining "what to do" at each stage of the task. Due to the length constraints of the paper, please refer to Appendix A for detailed implementation.

## 3.3 Motion Key-state Identification

While the SDM effectively recognizes semantic key states, they primarily outline the task objectives, without delving into manipulation details required for transition to these target states. To enhance the grasp of mechanisms of "how to do" the task, we propose the Motion Capture Module (MCM) to identify the motion key states within the interval defined by two semantic key states.

Specifically, we measure the intensity of motion between two consecutive states by the absolute value of optical flow of their corresponding observations, which is captured by the Farneback [33] optical flow estimation algorithm $F$. The $p$-th motion key state represents the critical transition from semantic key state $s_{I_{p-1}^s}$ to $s_{I_p^s}$, whose index $I_p^m \in (I_{p-1}^s, I_p^s)$ is identified by following formula:

$$I_p^m = \arg \max_{j \in (I_{p-1}^s, I_p^s)} |F(o_{j-1}, o_j)|. \tag{5}$$

By identifying the most intense optical flow, we select the related states as the motion key states, with index set $I^m$, which indicate "how to do" the target task.

## 3.4 Importance Weight Adaption

Based on the extracted hybrid semantic key states and motion key states, we develop the Importance Weight Adaption method to guide the agent in online exploration. Previous studies [15] typically assumed that the importance weights assigned to each state in expert trajectory $\nu$ in Equation 3 are equal, namely, $\nu = \frac{1}{N^d}\mathbf{1}$. While it aids trajectory-matching, the estimated reward under this assumption is disarmed of task awareness. In this module, we enhance task-aware reward by emphasizing the guidance of semantic and motion key states. Thus, we design the importance weight of expert demonstration, which was decomposed into $K$ subgoals, as a Gaussian Mixture Distribution $\mathcal{D}$:

$$\mathcal{D} = \frac{\sum_{i=1}^K A_i^s \mathcal{N}(I_i^s, \sigma_s^2) + \sum_{i=1}^K A_i^m \mathcal{N}(I_i^m, \sigma_m^2)}{\sum_{i=1}^K A_i^s + \sum_{i=1}^K A_i^m}, \tag{6}$$

where we adjust the semantic variance $\sigma_s$ to be smaller than the motion variance $\sigma_m$, guiding the agent to pay more attention to semantic information. The weights of these two types of key states $A_i^s$ and $A_i^m$ are introduced to ensure the dominance of task completion status when reward estimation. The choice of hyperparameters is detailed in Appendix B.2

Additionally, we replace the uniform distribution assumption of expert demonstrations $\nu$, with the normalized distribution $\mathcal{D}$, enhancing task-aware exploration during online imitation learning.

### 3.5 Learning Paradigm

To fully utilize the expert demonstrations, we employ an offline-to-online methodology, initially training a behavior cloning policy $\pi^b$ using the state-action pair $(s_d, a_d)$ in expert trajectory $T^d$:

$$L_{BC} = \mathbb{E}_{(s_d, a_d) \sim T^d} \left\| a_d - \pi^b(s_d) \right\|^2. \tag{7}$$

Initialized our exploration policy $\pi^e$ as the behavior cloning policy $\pi^b$, the agent could explore the environments with task-aware knowledge and measure the similarity between exploration trajectory and expert demonstrations. Further, we utilize the n-step Deep Deterministic Policy Gradient (DDPG) [34] with our estimated exploration reward $r^e$ to update our policy $\pi^e$, and Q-function $Q_\theta$:

$$\pi^e = \arg \max_\pi \left[ (1 - \lambda(\pi)) \mathbb{E}_{(s_e, a_e) \sim T^e}[Q_\theta(s_e, a_e)] - \alpha \lambda(\pi^e) \mathbb{E}_{(s_d, a_d) \sim T^d} \left\| a_d - \pi(s_d) \right\| \right]. \tag{8}$$

In this step, we employ the adaptive function $\lambda$ following the Regularized Optimal Transport (ROT) [15] to effectively balance the weight between BC regularization and online exploration:

$$\lambda(\pi^e) = \mathbb{E}_{(s, \cdot) \sim T^e} \left[ \mathbf{1}_{Q_\theta(s, \pi^b(s)) > Q_\theta(s, \pi^e(s))} \right]. \tag{9}$$

However, the limited expert demonstrations commonly fail to offer task-aware reward $r^e$ for extensive online exploration space. The KOI method we proposed release this issue via the guidance of hybrid semantic key states and motion key states, accelerating the online imitation learning.

## 4 Experiments

### 4.1 Experiments Setting

To comprehensively evaluate our key-state guided method, we initially conducted experiments across 6 tasks from the Meta-World suite [35] to validate the efficacy of our method. Further, we assess KOI's performance in more complex manipulation tasks, we conduct 3 tasks from the LIBERO suite [36]—notable for its complex scenarios, large action spaces, and long task sequences.

In offline imitation phase, we collected 1 demonstration and 50 demonstrations per task in the Meta-World suite and LIBERO suite to train BC policy, respectively. For online imitation phase, we loaded pretrained BC policy and initialized critic network. Details of settings are attached at Appendix B.

### 4.2 Comparison Experiments

To evaluate our methods, we conducted comparison experiments with various methods, all methods are initialized with pretrained behavior cloning policy $\pi^b$ to ensure a fair comparison.:

- BC represents the pretrained behavior cloning policy $\pi^b$.
- UVD [37] utilize pretrained visual encoder R3M [38] for task decomposition in image-goal reinforcement learning.
- RoboCLIP [25] leverages the pretrained video encoder S3D [39] to estimate the task-aware reward function instead of the reward of each state.
- ROT [15] employs Optimal Transport to estimate the distance between exploration trajectory and expert demonstrations, which focus on fine-grained trajectory matching.

As the results presented in Figure 2 shown, the UVD method, which emphasizes semantic subgoals rather than the mechanisms of "how to do", often fails to accomplish the target task. Similarly, RoboCLIP leverages a video encoder to capture trajectory motion, but the absence of per-state reward hinders its ability to guide online imitation. Although ROT employs fine-grained reward and shows commendable performance, it sadly neglects task-aware reward, thereby impeding efficient exploration. This issue is highlighted in the qualitative analysis detailed in Section 4.4.

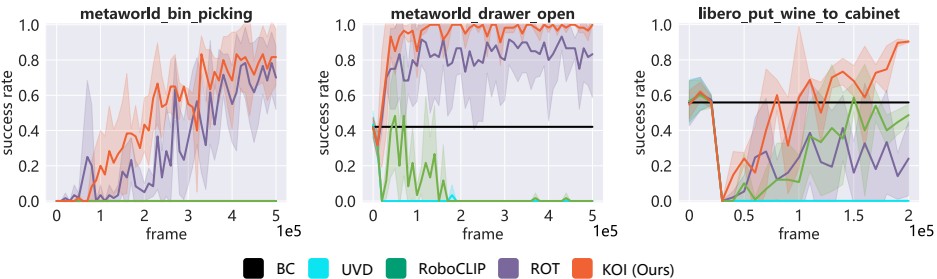

Figure 2: A subset of experiment results on Meta-World and LIBERO suites, with exploration 5×1e5 and 2×1e5 timesteps respectively. The shaded region represents ± 1 standard deviation across 3 seeds. The results prove that our KOI method excels in sample efficiency compared to others.

In contrast, our KOI method achieves more efficient online imitation learning in simple Meta-World tasks such as "drawer open". For the challenging "bin picking" task, characterized by multiple manipulation subgoals, only the fine-grained reward estimation methods (our method and ROT) have achieved success. Furthermore, our method outperforms ROT by leveraging a thorough comprehension of both semantic and motion key states, providing a distinct advantage.

The tasks in the LIBERO suite present increased difficulties, due to longer task sequences and larger action spaces. The results demonstrate that KOI not only yields more efficient learning but also maintains greater stability, benefiting from the comprehensive guidance from the hybrid key states.

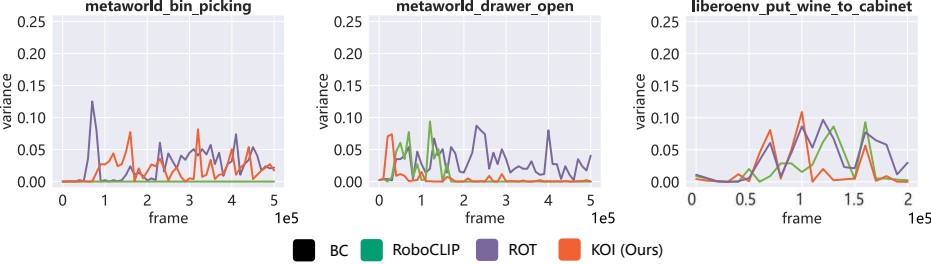

Figure 3: The trend of variance during online imitation learning. The variances of KOI continuously decreases over time.

We further discuss the trend of variance over time. As shown in Figure 3, the variance of KOI decreases continuously during exploration while the performance increases, which indicates that our method can enhance model performance across a diverse range of environments, thereby demonstrating its robustness and stability.

### 4.3 Ablation Experiments

Our method proposes the Semantic Decomposition Module (SDM) and Motion Capture Module (MCM) to provide task-aware knowledge for trajectory-matching reward estimation. To evaluate the effectiveness of each module, we conducted ablation experiment with the following three settings:

- w/o. Both: This setting selects the state every 10 timesteps as the key state and utilizes them to adjust the importance weight of expert demonstrations.

- w/o. MCM: This setting only uses states selected by visual-language models to adjust the importance weight of expert demonstrations.

- w/o. SDM: This setting removes the semantic subgoals inferred by visual-language models and selects motion key states within every 10-state intervals.

As shown in Figure 4, the absence of either semantic or motion key states detrimentally affects the efficiency of online exploration, particularly in the complex "bin picking" manipulation task. No-

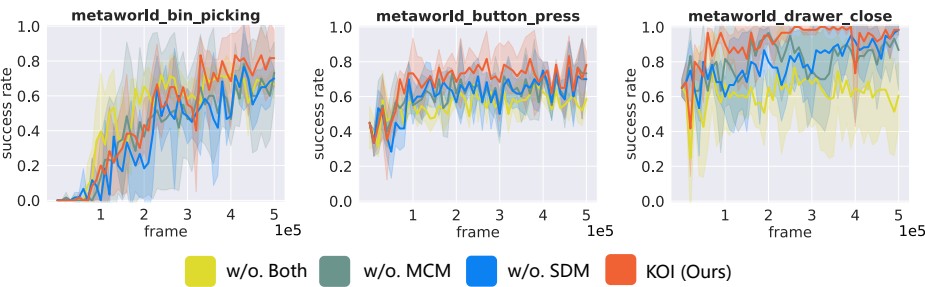

Figure 4: Ablation experiments of our proposed Semantic Decomposition Module (SDM) and Motion Capture Module (MCM). The shaded region represents ± 1 standard deviation across 3 seeds.

tably, both the Semantic Decomposition Module and Motion Capture Module we proposed surpass "w/o. Both" setting in most tasks, demonstrating the effectiveness of each module.

Besides, the performance of the "w/o. MCM" setting is comparable to the "w/o. SDM" setting, demonstrating the equal importance of both types of key states. Additionally, the shaded region for the "w/o. MCM" setting is larger than the other methods, indicating that motion key states contribute to stabilizing online exploration by providing detailed guidance on "how to do" tasks.

### 4.4 Qualitative Analysis

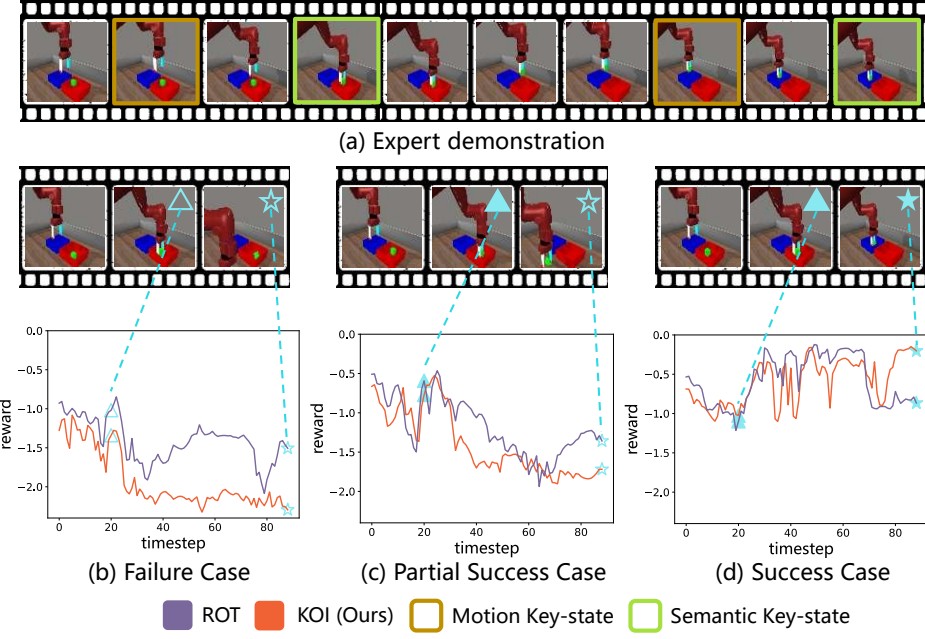

Figure 5: The qualitative analysis of our method. (a) demonstrates the selected semantic and motion key states. (b)-(d) illustrates three representative cases and estimated reward values in "bin picking" tasks. The triangular and pentagonal icons represent two objectives of this task, with the filling denoting completion status, the corresponding estimated rewards are linked by dashed lines.

We visualize the semantic and motion key states selected by our method. As shown in Figure 5(a), our semantic key states effectively extract subgoals of the task. Our motion key states, capture the most intensive optical flow between semantic subgoals, allowing agents to grasp motion dynamics.

Furthermore, we present three representative cases in the "bin picking" task. As shown in Figure 5(b), when the agent fails to complete any subgoals, our method estimates a lower reward to

discourage unsuccessful exploration. In cases where the agent completes only the first subgoal of Figure 5(c), the estimated reward is high during the initial stage but lower at the final stage. In contrast, ROT incorrectly estimates a high reward for the final stage. The last case, shown in Figure 5(d), demonstrates that our method provides a high reward estimation when all subgoals of this task are completed, thereby facilitating effective online exploration.

## 4.5 Real-world Results

As shown in Table 1, We conduct real-world experiments using an XARM robot equipped with a Robotiq gripper. To comprehensively evaluate the performance, the entire imitation learning is divided into three stages: offline, online, and test. For each task, we collect 10 human expert demonstrations to train the Behavior Cloning (BC) policy during the offline phase and use only 1 expert demonstration to estimate reward during the online phase. All the evaluations are conducted at varying initial object positions. Implement details can be found in Appendix C.

We conduct comparative experiments between our method and previous trajectory-matching method [15], reporting the success rates of each phase in Table 1. As the results show, the ROT method commonly struggle to provide effective reward estimation for unseen initial positions. However, our method utilizes task-aware information to improve the reward estimation, thereby enhancing the agent's exploration efficiency and performance. The differences are most significant in the task "put the cube on the scale", which requires the highest precision in manipulation, making it difficult to complete the task by simply matching trajectories with expert demonstrations.

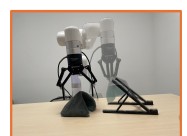 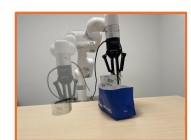 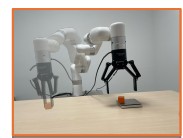

| Method | Take the rag off | | | Put the tape in the box | | | Put the cube on the scale | | |
|---|---|---|---|---|---|---|---|---|---|
| | Offline | Online | Test | Offline | Online | Test | Offline | Online | Test |
| BC | 50% | – | – | 40% | – | – | 30% | – | – |
| ROT | 50% | 60% | 70% | 40% | 45% | 60% | 30% | 20% | 30% |
| Ours | 50% | **70%** | **100%** | 40% | **70%** | **90%** | 30% | **40%** | **50%** |

Table 1: The demonstrations and comparative results of our 3 real-world robotic manipulation tasks. The initial state of each task is represented in a lighter shade, while the completed state is depicted in a darker shade. The results indicate that our method achieves more efficient exploration and better overall performance.

## 5  Conclusion and Limitation

In this work, we propose the hybrid Key-state guided Online Imitation (KOI) learning method, which extracts the semantic and motion key states for trajectory-matching reward estimation for online imitation learning. By decomposing the target task into the objectives of "what to do" and the mechanisms of "how to do", we refine the trajectory-matching reward estimation to encourage task-aware exploration for efficient online imitation learning. The results from simulation environments and real-world scenarios prove the efficiency of our method.

**Limitations.** The online imitation learning in this study initializes an critic, as shown in our results, which negatively impacts early exploration. In future work, offline reinforcement learning could be employed to pretrain both the actor and critic, potentially enhancing initial online learning.

## ACKNOWLEDGEMENT

This work is supported by the National Natural Science Foundation of China (NO.62106272), the National Natural Science Foundation of China (62376222), Shanghai AI Laboratory, National Key R&D Program of China (2022ZD0160101), and Young Elite Scientists Sponsorship Program by CAST (2023QNRC001).

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

# Appendix

## A    Details of Semantic Decomposition Module

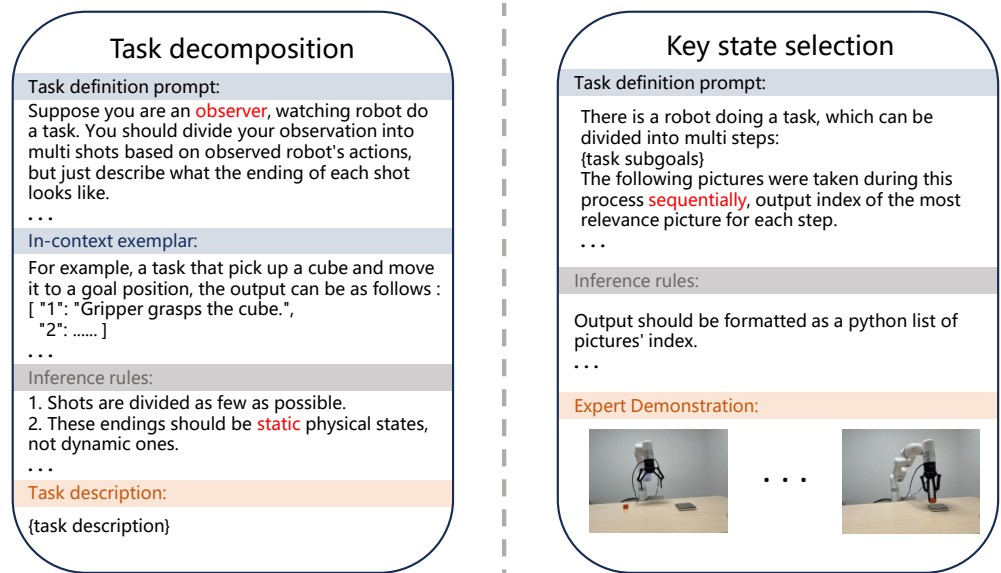

Figure 6: The pre-defined prompt used in our Semantic Decomposition Module. Some keywords can enhance the quality of decomposition and selection.

In the Semantic Decomposition Module (SDM), we employ GPT-4v to extract the semantic key states from expert demonstrations, whose rich world knowledge and strong generalization capabilities have been proven in various visual-language tasks [31, 32]. The process of extraction includes two stages: task decomposition and key state selection.

The first stage takes the description of the task as input, decomposing the manipulation process into multiple subgoals. Concretely, we use the name of the task as the task description. These subgoals will be the criteria for selecting key states from expert demonstrations.

The key state selection stage would take the observation queries and subgoal descriptions into GPT-4v, and get the key state indexs. The prompts for the two stages are shown in Figure 6. Due to the API and context length restrictions, we select states from an expert demonstration every $T$ step to form the query expert observation sets. In our experiment, $T$ is assigned as 10. Empirically, we found that there will be chronological confusion if only one subgoal is passed in each query, *e.g.*, selecting the fifth frame for the first subgoal while the third frame for the second subgoal. Passing all subgoals in one go solves this problem effectively.

## B    Details of Experiments

### B.1    Experiments setting

As illustrated in Section 4, the LIBERO suite [36] is notable for complex scenarios, large action spaces, and long task sequences. Thus, to promote policy learning, the policy in the LIBERO suite utilizes both image and proprioceptive as input, while the policy in the Meta-World suite only utilizes image as input. Besides, 50 expert demonstrations are borrowed from their open-source release for BC training in LIBERO suite, while the estimation of the agent's exploration reward is only based on 5 of them, which reduces the time cost of computation reward. In addition, when the task is finished, each state of the successful exploration will gain a task-finish reward, to promote policy learning. More details of environment setting can be found at Table 2.

During imitation learning phase, each task was trained using a batch size of 256 over 50k epochs. For the online imitation learning phase, we loaded the pretrained BC policy and initialized the critic network. The model then interacted with the environment over 500k timesteps in the Meta-World suite and 200k timesteps in the LIBERO suite, allowing us to observe the efficiency of different online imitation learning methods. For a fair comparison to ROT [15], we train the policy using a stack of 3 consecutive RGB frames in Meta-World suite, and each action in the environment is repeated 2 times.

| Suite | Parameter | Value |
|---|---|---|
| Meta-World | Use proprioceptive | False |
| | Image size | $84 \times 84$ |
| | Action shape | 4 |
| | Frame stack | 3 |
| | Action repeat | 2 |
| | Seed frames | 12000 |
| | Task finish reward | 0 |
| | Demonstration(s) for BC | 1 |
| | Demonstration(s) for online finetuning | 1 |
| LIBERO | Use proprioceptive | True |
| | Image size | $128 \times 128$ |
| | Action shape | 7 |
| | Frame stack | 1 |
| | Action repeat | 1 |
| | Seed frames | 24000 |
| | Task finish reward | 5 |
| | Demonstration(s) for BC | 50 |
| | Demonstration(s) for online finetuning | 5 |

Table 2: Details of environment settings.

## B.2   Hyperparameters

The complete list of hyperparameters is provided in Table 3. As shown, all the methods differ only in reward estimation, *i.e.*, components like encoder and RL backbone are the same. When modeling the importance weight of expert demonstration, we apply distinct weights and standard deviations for semantic and motion key states, respectively. The weight assigned to the ultimate goal of the task is set higher than that of the subgoals, thereby enhancing the agent's awareness of task completion.

## B.3   Results of Comparison Experiments

In addition to the results provided in Section 4.2, Figure 7 shows the performance of KOI on all tasks – 6 from the Meta-World suite [35] and 3 from the LIBERO suite [36]. On all tasks, KOI has shown its sample efficiency compared to other reward estimation methods [15, 37, 25]. However, the "door unlock" task, as shown in Figure 12, requires the robot arm to rotate the handle to unlock the door. However, during online exploration, the agent could complete this task by using its body instead of its gripper to rotate the handle, due to the imprecise physical simulation. The tricky policy can complete the task without imitating the expert's trajectory, resulting in the degradation

| Method | Parameter | Value |
|:---:|:---:|:---:|
| Common | Replay buffer size | 150000 |
| | Learning rate | $1e^{-4}$ |
| | Discount $\gamma$ | 0.99 |
| | $n$-step returns | 3 |
| | Mini-batch size | 256 |
| | Agent update frequency | 2 |
| | Critic soft-update rate | 0.01 |
| | Feature dim | 50 |
| | Hidden dim | 1024 |
| | Optimizer | Adam |
| | Exploration steps | 0 |
| | DDPG exploration schedule | 0.1 |
| | Target feature processor update frequency(steps) | 20000 |
| | Reward scale factor | 10 |
| | Fixed weight $\alpha$ | 0.03 |
| | Linear decay schedule for $\lambda(\pi)$ | linear(1,0.1,20000) |
| KOI | Weight of semantic key states $A_i^s$ ($i < N^s - 1$) | 0.15 |
| | Weight of the last semantic key state $A_{N^s}^s$ | 0.35 |
| | Weight of motion key states $A^m$ | 0.05 |
| | Standard deviation of semantic key states $\sigma_1$ | 10 |
| | Standard deviation of motion key states $\sigma_2$ | 25 |

Table 3: List of hyperparameters.

of performance during online finetuning with any reward estimation method as compared to the pretrained policy.

## B.4  Ablation of Semantic Decomposition Module

We provide the ablation study of the Semantic Decomposition Module, which select the semantic key states with different methods:

- Human Annotation: We manually annotate the key states in the expert demonstration.

- UVD_R3M: Utilizing the pretrained R3M visual model as a visual encoder for task decomposition, we select semantic key states to adjust the importance weights in the expert demonstration.

- UVD_trained: We utilize the visual encoder from the Behavior Cloning model, which is trained on the expert demonstrations, to select semantic key states to adjust the importance weights of the expert demonstration.

The results in Figure 8 demonstrate that our model, designed based on GPT-4v, can fully leverage the comprehension capabilities of large multimodal models, thereby achieving better decomposition effects than pre-trained models and trained sequential models. Additionally, our model achieves

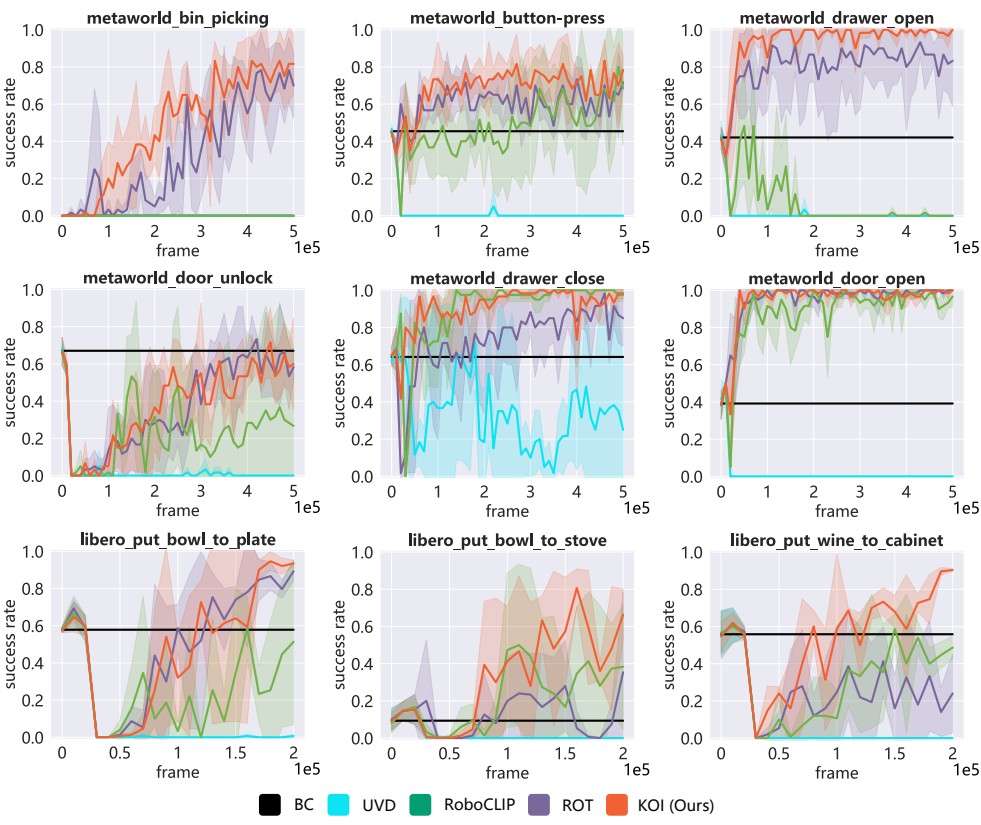

Figure 7: The experiment results on 6 tasks from Meta-World environments and 3 tasks from LIBERO environments. The shaded region represents ± 1 standard deviation across 3 seeds.

results comparable to manually selected key frames, while reducing labor costs and enabling easier scalability in future processes.

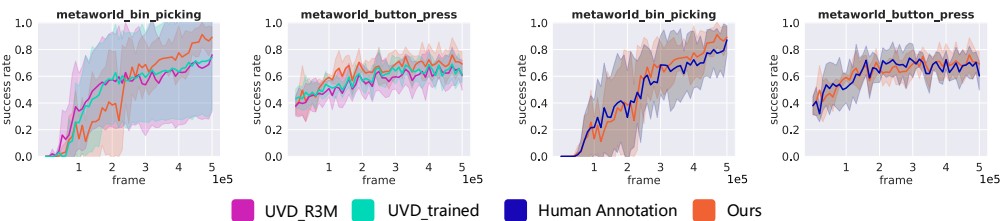

Figure 8: The ablation experiment of Semantic Decomposition Module.

## B.5 Ablation of Motion Capture Module

We provide the ablation study of the Motion Capture Module (MCM), which selects the motion key states with different methods:

- w/o. MCM: This setting only uses states selected by visual-language models to adjust the importance weight of expert demonstrations.
- uniform Motion Capture: This setting selects the motion key states every 5 steps within the interval defined by two semantic key states.

As shown in Figure 9, the uniform motion selection method would choose noisy keyframes, affecting the reward estimation. However, our motion capture module can extract useful information to facilitate online exploration.

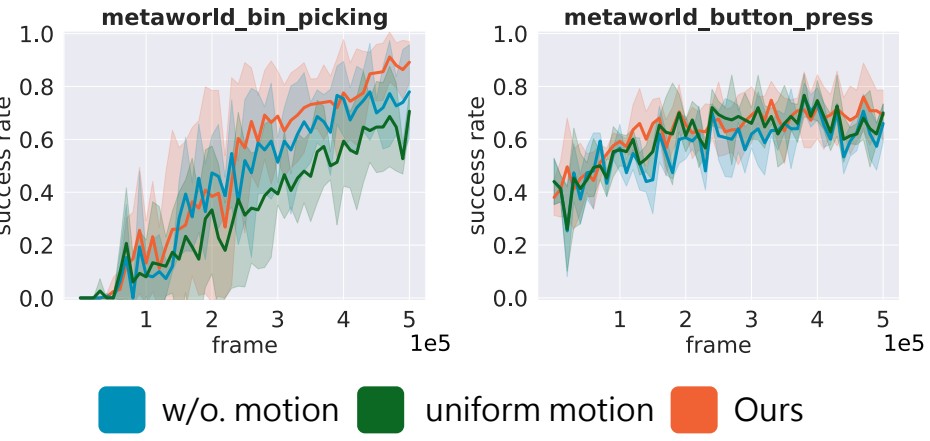

Figure 9: The ablation experiment of Motion Capture Module.

## B.6 Ablation of Both Modules

We also provide the ablation study of the Both Module:

- w/o. Both: This setting selects the state every 10 timesteps as the key state and utilizes them to adjust the importance weight of expert demonstrations.
- w/o. MCM: This setting only uses states selected by visual-language models to adjust the importance weight of expert demonstrations.
- w/o. SDM: This setting removes the semantic subgoals inferred by visual-language models and selects motion key states within every 10-state intervals.

As shown in Figure 10, the absence of either semantic or motion key states detrimentally affects the efficiency of online exploration, particularly in the complex "bin picking" manipulation task.

Besides, the performance of the "w/o. MCM" setting is comparable to the "w/o. SDM" setting, demonstrating the equal importance of both types of key states.

Additionally, the shaded region for the "w/o. MCM" setting is larger than the other methods, indicating that motion key states contribute to stabilizing online exploration by providing detailed guidance on "how to do" tasks.

Notably, both the Semantic Decomposition Module and Motion Capture Module we proposed surpass "w/o. Both" setting in most tasks, demonstrating the effectiveness of each module.

## C Real-World Experiments

In this work, we conduct 3 real-world robotic manipulation tasks, as shown in Figure 1. We conduct our real-world experiments using an XARM robot equipped with a Robotiq gripper. To facilitate efficient policy learning, we limit the robot's action space to 4 dimensions (x, y, z, gripper) and utilize both image and proprioceptive inputs for the policy. To ensure the safety of the exploration, we restrict the robot arm's gripper to operate within a certain safety range.

For each task, we collect 10 human expert demonstrations to train the Behavior Cloning (BC) policy during the offline phase and use a single expert demonstration to estimate online imitation reward

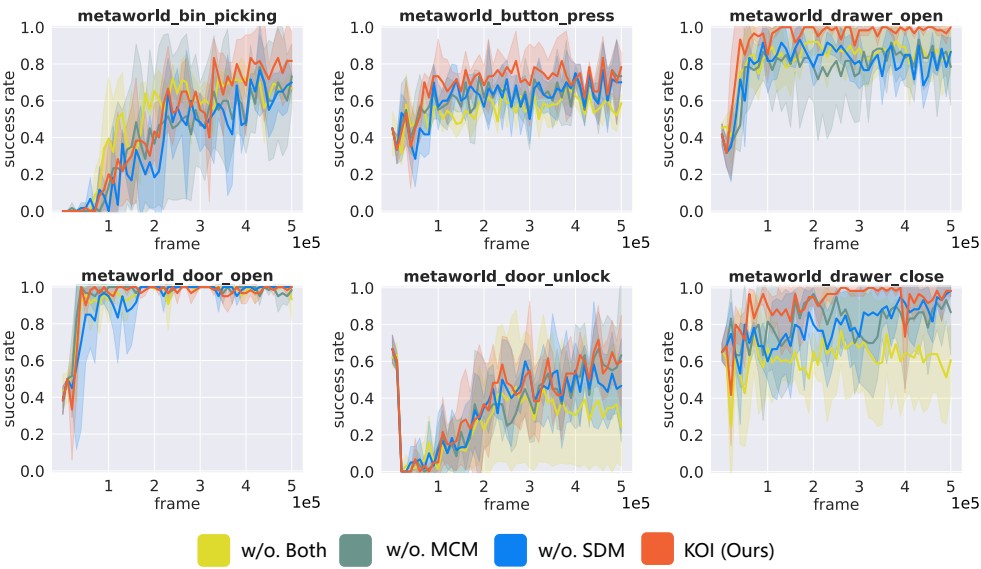

Figure 10: The ablation experiment of both modules.

during the online phase. To evaluate the pretrained BC policy, we conduct 10 experiments at varying initial object positions to calculate the average success rate. We then conduct 20 times of online explorations, recording their average success rates. Finally, we load the updated policy and conduct 10 offline tests to assess its performance.

**Offline:** For image data, we resize them to $120 \times 160$ and extract 128-dimensional representations through a shallow CNN network. For proprioceptive data, we collect the 6D pose information of the robot arm's end-effector, the 7-axis degrees of freedom joint information, and the gripper's angle. We concatenate the proprioceptive and then map them to the 128-dimensional representation through a fully connected layer. We train the BC policy $\pi^{bc}$ with 10 expert demonstrations over 200 epochs and evaluate the performance at 10 various initial object positions.

**Online:** We load the pretrained policy $\pi^{bc}$ and interact with environments within 20 times during online exploration. To ensure the safety of the robot, we restrict the operation range of the gripper to prevent it from colliding with the table.

As mentioned in the limitation, the mismatch between pretrained BC policy and initial critic would negatively impact early exploration learning. Consequently, to ensure both safety and efficiency during real-world exploration, we substitute the adaptive function $\lambda$ in Equation 8 with a constant value of 0.9 and provide a sparse reward to indicate whether the task has been completed. Through this modification, we enable the robot to explore the environment stably and develop the exploration policy $\pi^e$.

To ensure the fairness of the experiments, we also set the adaptive weights of ROT in the comparative experiments as the constant value of 0.9.

**Test:** We load the updated exploration policy $\pi^e$ to evaluate the performance at 10 various initial object positions.

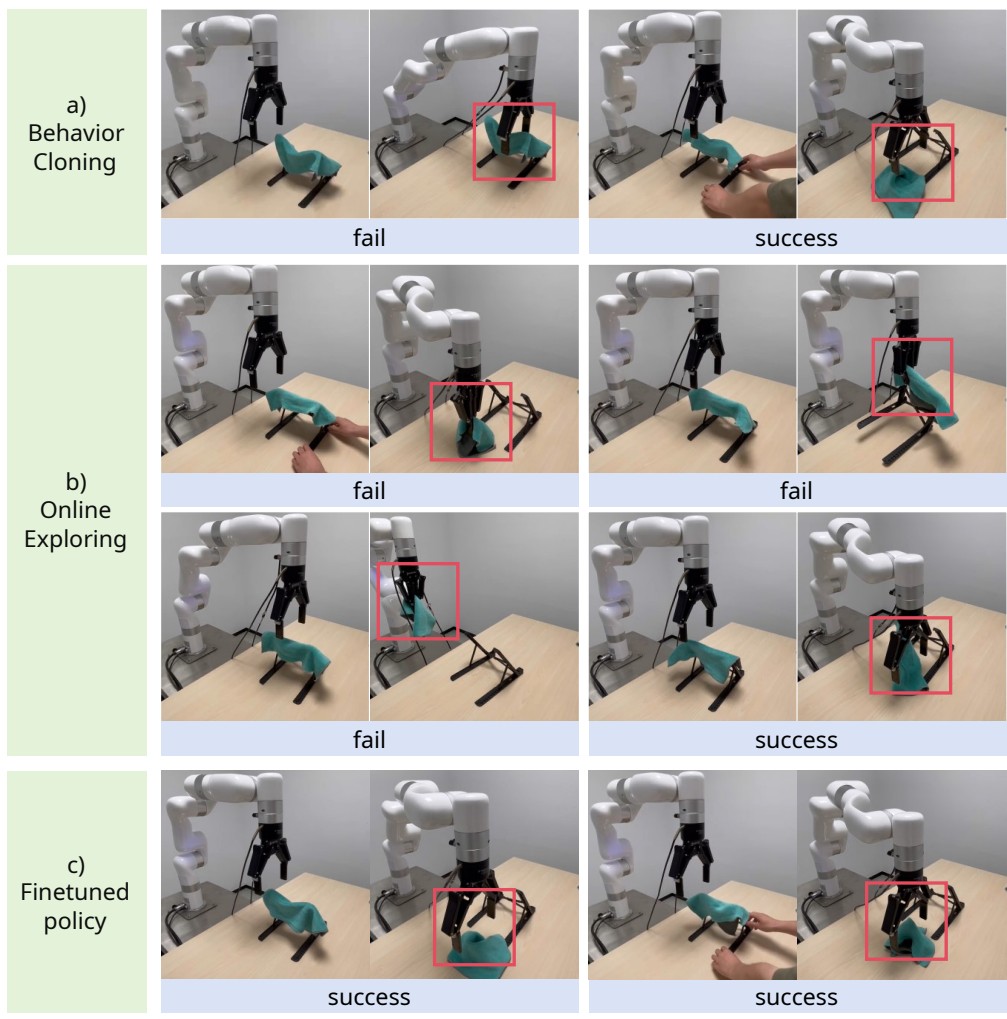

Figure 11: Overview of our real-world experiment on "take the rag off" task. KOI significantly accelerates the online finetuning, guiding the agent to complete the task even when objects are placed in different positions.

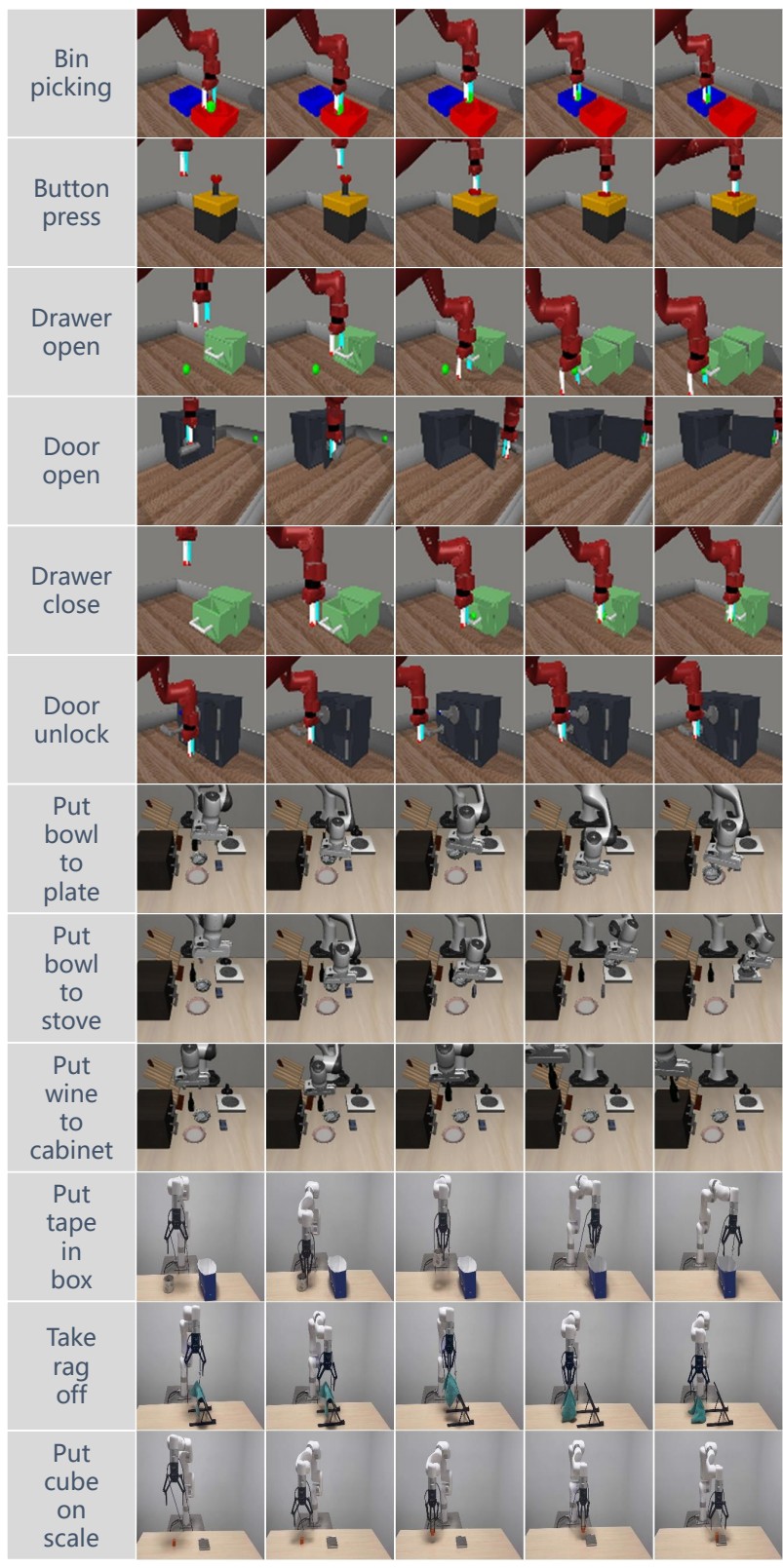

Figure 12: Example trajectories for 6 tasks from Meta-World suite, 3 tasks from LIBERO suite, and 3 real robot tasks, sequentially.

