# OpenReview forum: "KOI: Accelerating Online Imitation Learning via Hybrid Key-state Guidance"
_robot-learning.org/CoRL/2024/Conference — CoRL 2024_

### Official Review · Reviewer_3bAV · 2024-07-14
**key insight is to facilitate reward estimation for online imitation learning using task relevant key state information - particularly when there are limited demonstrations available**

**Originality:** 3
**Technical Quality:** 3
**Clarity Of Presentation:** 3
**Potential Impact:** 2
**Recommendation:** 4
**Confidence:** 4

**Review:**

Quality:
The paper is well-structured and presents a clear methodology for improving reward estimation from limited demonstrations for online imitation learning.

Clarity:
The paper is generally clear and easy to follow. The authors provide a detailed explanation of the problem, the proposed solution, and the results. The use of diagrams and qualitative analyses aids in understanding the methodology and its benefits. However, some sections, such as the mathematical formulations, could benefit from additional explanations to enhance clarity for readers less familiar with the concepts. There are some outputs that are not clearly identified eg what is the representation of I^s and I^m  - please clearly define all the terms, their dimensionality and how they are used to derive the reward. I can see that you utilise these terms to define a normal distribution but and better justification on the choice of sigma here is required. How would this extend to new tasks the choice of these variables - how would one estimate them for a new task?

Why does UVD fail across all tasks? Why not use it to identify the keypoint states instead if GPT4? - seems like a more relevant use of this method than direct comparison to online imitation learning which it was not designed for.

In figure 3 it would be good to see the graph without both systems as well.

Originality:
The approach of integrating semantic and motion key states for reward estimation in online imitation learning is novel. The use of visual-language models to decompose tasks and optical flow to capture motion dynamics adds a unique dimension to the research. This combination of techniques is innovative and I do like how the authors combined the use of foundation models in a unique manner to improve reward estimation.

Significance:
I am not full convinced on the significance of the results. The major claim is it improves sample efficiency in downstream exploration however based on the key results it seems like ROT and the KOI method are converging at similar times across the majority of the experiments. While KOI does get slightly higher reward I am not convinced why this is a result of the task-dependent key state information for reward estimation - an explanation of why this is should be included. Furthermore, the results are missing a key component/experimental results demonstrating how the number of demonstrations used for reward estimation affects downstream task performance with the proposed semantic key points and without them. Basically, what is 'limited' expert demonstration quantitatively - and at what point is limited no longer relevant that we don't need this semantic decomposition to facilitate reward estimation?

Strengths:

Novel Approach: The integration of semantic and motion key states for task decomposition is innovative and effective.
Comprehensive Evaluation: The method is validated through extensive experiments in both simulated and real-world environments.
Practical Applicability: Demonstrates success in real-world robotic tasks, indicating the method's robustness and applicability.
Clear Methodology: The step-by-step explanation and visual aids enhance the understanding of the proposed approach.

Weaknesses:

Clarity of formulation: Some sections could benefit from additional explanations and definitions to improve clarity as discussed above.
Results to demonstrate the key claim of limited demonstrations and how many is 'limited' are missing.
How does the performance of the SDM and MCM impact performance - how robust is this approach if these modules decomposed incorrectly? Will it only work on simple tasks where it is straightforward to identify these key states?
You empirically chose 10 for T. What impact does this have on the system performance?
There seem to be many hyperparameters that are not ablated to understand their impact on the system.

Refer to the questions raised below.

**Quality Of The Limitations Section:**

1

**Questions For Rebuttal:**

1. Can you provide more detailed explanations or examples for the mathematical formulations, especially the OT-based reward estimation and the Importance Weight Adaption?
2. The paper mentions a reliance on pretrained behavior cloning policies. How significant is this dependency, and what would be the potential impact on early exploration efficiency without such pretraining?
3. Could you provide more insights or results regarding the choice of variance parameters \( \sigma_1 \) and \( \sigma_2 \) for the Gaussian Mixture Model? How sensitive is the model's performance to these parameters?
4. Can you give more detailed descriptions of the real-world experiments, including the setup, specific tasks, and any encountered challenges?
5. How does the KOI method scale with more complex tasks? Any additional insights or experiments on generalization capabilities would be valuable.
6. How does the semantic decomposition from GPT-4 impact performance. What is the variation in this process? Would something like UVD be more suitable to extend to more complex tasks where the semantic keypoint starts arent as obvious to a vision model? Explain how reliant/robust your method is to the selection of these keystates and comment comment on how general it is to scale to arbitrary tasks.
7. In figure 3, could you include a graph without both the Semantic Decomposition Module (SDM) and the Motion Capture Module (MCM) to better illustrate their individual contributions?
8. Can you clarify the representation, dimensionality, and usage of \( I^s \) and \( I^m \)? How are these terms specifically utilized to derive the reward, and what is the justification for the choice of sigma values?
9. Can you explain why the KOI method shows slightly higher rewards compared to ROT, and how this result is attributed to task-dependent key state information for reward estimation?
10. Could you provide experimental results demonstrating how the number of demonstrations used for reward estimation affects downstream task performance with and without the proposed semantic keypoints? What is considered 'limited' in terms of expert demonstrations?

**Robotics Focus:**

4

**Summary Of Paper:**

The paper presents a novel approach called Hybrid Key-state guided Online Imitation (KOI) learning to enhance the efficiency of Online Imitation Learning (OIL). The main idea is to address the challenges of task-aware reward estimation by decomposing tasks into "what to do" (semantic key states) and "how to do" (motion key states). By integrating these key states, the method refines reward computation, encouraging more efficient online exploration. The KOI method leverages visual-language models and optical flow for key state extraction, which is validated through experiments in simulated environments (Meta-World and LIBERO) and real-world robotic manipulation tasks, demonstrating improved sample efficiency and practical applicability.

**Summary Of Recommendation:**

I believe this is good idea and I do appreciate the unique use of higher level reasoning to faciliate reward estimation. There are some components that are not clear within the paper to fully convince me of the key contributions and impact of this approach on the online imitation learning setting and how this method can scale/generalise to other tasks.

---

### Official Review · Reviewer_raZY · 2024-07-23
**Simple but effective improvement over an existing baseline on Online Imitation Learning**

**Originality:** 3
**Technical Quality:** 3
**Clarity Of Presentation:** 4
**Potential Impact:** 3
**Recommendation:** 3
**Confidence:** 3

**Review:**

These are the strengths and weakness of the paper in my opinion.

Strengths
1. Simple but intuitive idea of utilizing task decomposition of expert trajectories for better reward estimation.
2. Clever use of recent advances in sequential models (vision-language models) to automate this task decomposition task.
3. Convincing performance improvements over valid baselines on Online Imitation Learning.
4. The method's superiority is further emphasized with qualitative and empirical experiments.

Weakness
1. Incremental advancement over an existing algorithm.
2. The algorithm is limited by the generalization capability of vision-language foundation models. There might be many instances beyond the said "naturally" hierarchical manipulation tasks, where the model might fail to decompose the task effectively.
3. I doubt if the task decomposition achieved by a foundation model on the expert trajectories would outperform a human expert annotating the trajectories with key states. If the expert trajectories can be collected by an expert, it can as well be labelled into subtasks by the same expert or another human.

**Quality Of The Limitations Section:**

2

**Questions For Rebuttal:**

1. Is the use of pre-trained vision-language model the only way to achieve this task decomposition. Can a sequential model be explicitly trained to decompose the task in case the pre-trained model fails at this task?

2. What would happen if the tasks are not inherently hierarchical? Let's say the simpler tasks like a cheetah run from dmc, as performed in the original ROT paper?

3. Can't we let a human do the task decomposition instead of the GPT model?

**Robotics Focus:**

4

**Summary Of Paper:**

The paper proposes a simple modification to an exisiting online Imitation Learning algorithm, by incorporating task decomposition and subsequent estimation of task aware rewards using Optimal Transport. The task decomposition is achieved by leveraging exisitng vision language models and optical flow etimation techniques. The experiments show how the proposed modification with task decomposition as an inductive bias for learning give superior quantitative and qualitative results over valid baselines.

**Summary Of Recommendation:**

I recommend for a Weak Accept with a lower confidence.

---

### Official Review · Reviewer_APBJ · 2024-07-25
**Paper applying abstraction to imitation learning**

**Originality:** 3
**Technical Quality:** 3
**Clarity Of Presentation:** 2
**Potential Impact:** 3
**Recommendation:** 3
**Confidence:** 3

**Review:**

## Summary

This paper introduces the hybrid Key-state guided Online Imitation (KOI) learning approach, which aims to improve online imitation learning by addressing the gap between extensive online exploration space and limited expert trajectories. The authors propose decomposing tasks into "what to do" (objectives) and "how to do" (mechanisms) to estimate more precise task-aware imitation rewards for efficient online exploration.
The KOI method consists of three main components:

* Semantic Decomposition Module (SDM): Uses GPT-4v to decompose task descriptions into subgoals and identify corresponding key states.
* Motion Capture Module (MCM): Employs optical flow to capture motion key states between semantic key states.
* Importance Weight Adaptation: Adjusts the importance weights of expert demonstration states using a Gaussian Mixture Distribution based on the extracted key states.

The authors evaluate their method on 6 Meta-world tasks and 3 LIBERO tasks, comparing it to baseline methods such as Behavior Cloning (BC), ROT, RoboCLIP, and UVD. They also conduct ablation studies and real-world experiments using an Xarm robot.

## Strengths

* Novelty: the integration of semantic and motion key states for guiding online imitation learning is an interesting idea, while abstraction is not new, I think this approach is sufficiently novel

* Evaluation: the authors do a good job of evaluating their method across a range of simulated tasks and demonstrating transfer to a real robot system

* Qualitative analysis: I appreciated the author's qualitative discussion of the method and intuition for what is learned

## Weaknesses

* Unclear justification for design choices: Several key design choices are not well-motivated or evaluated. In particular, the way that motion key states are determined was not motivated or evaluated. I also had trouble understanding the motivation behind the method for importance weight adaptation

* Limited real world evaluation: while there is some evaluation on a real robot, it is a bit limited. It's not clear that a reader could fully reproduce the experiment. Similarly, additional tasks on the real robot would help to argue for the practicality of the method.

* Lack of comparison to other hierarchical or subgoal-based methods: for a paper like this, I expect the authors to experiment with and compare to other imitation learning methods that leverage abstraction/hierarchy.

## Typos and clarity issues

* Line 132: "To extract the important motion key states index set I_m = {I_m_1, .., I_m_j, .., I_m_K}, we compute the optical flow between successive expert observation o_j and o_j+1, selecting the most intense state between two semantic key states, which represents a critical state transition during the manipulation process:" --- The sentence is long and could be split for clarity.

* Line 122: "To manage the number of input images, we select state every T step to form the query expert observation sets O." --- Should be "we select states every T steps to form..."

* Equation 5: The equation needs more explanation and context.

* Line 206: "w/o. SDM: This setting removes the semantic subgoals inferred by visual-language models and selects motion key states within every 10 states interval." Should be "10-state interval" or "10-state intervals"

Update after rebuttal: the authors' response largely addresses my concerns; I have updated my score accordingly.

**Quality Of The Limitations Section:**

2

**Questions For Rebuttal:**

It seems like the semantic key states are identified entirely from the language model. How would your approach work in a task where there are multiple plausible ways to accomplish it, but the expert demonstrations select one? Would you still learn effectively in this situation?

**Robotics Focus:**

4

**Summary Of Paper:**

This paper presents a novel approach called hybrid Key-state guided Online Imitation (KOI) learning to improve online imitation learning. The method addresses the challenge of efficiently exploring large state spaces with limited expert demonstrations by decomposing tasks into "what to do" (objectives) and "how to do" (mechanisms).

**Summary Of Recommendation:**

An interesting idea with room to improve on presentation and evaluation of the method

---

### Official Review · Reviewer_5oNU · 2024-07-26
**Review of submission 336**

**Originality:** 3
**Technical Quality:** 3
**Clarity Of Presentation:** 4
**Potential Impact:** 3
**Recommendation:** 4
**Confidence:** 4

**Review:**

# Clarity

The paper is generally well-written and easy to follow. The authors clearly articulate the motivation behind their work and provide a comprehensive overview of the problem they are addressing. The explanation of the KOI method, including the use of visual-language models and optical flow for extracting semantic and motion key states, is particularly well-structured. The figures and diagrams included in the paper are helpful in illustrating the proposed methodology and results. Overall, the clarity of the paper makes it accessible to a broad audience, including those who may not be experts in the field. Minor correction: In line 101 expert demonstration should be T^d and not T^e.

# Strengths

The paper presents an interesting and promising approach to improving online imitation learning by integrating visual-language models to extract semantically relevant information and combining this with optical flow to understand the necessary actions for task completion. This hybrid approach, which decomposes tasks into "what to do" and "how to do" components, is an interesting idea that leverages recent advances in multi-modal language models. The method shows promise in enhancing task-aware exploration and has potential applications in various domains requiring efficient learning from limited expert demonstrations.

# Weaknesses

The primary weakness of the paper lies in the evaluations provided by the authors. In Figure 2, there appears to be a significant amount of variance in the results. In several cases, the lower bound for KOI is lower than the average performance of baseline methods, particularly ROT and RoboCLIP. This raises questions about the consistency and reliability of the KOI method's performance. The authors should address the reasoning behind this high variability. Is it due to having only three trials? If so, I recommend conducting more trials to obtain more robust and reliable bounds for their experimental results. Additionally, the lack of a comparative study with the real-world robot arm is disappointing. The authors claim their method works in real-world settings but provide only a figure with the experimental setup and no detailed results or comparisons. This omission is particularly concerning given the variability observed in the simulation results. I recommend the authors perform a full comparative study using the same baselines with the real robot and include these findings in the paper to strengthen their claims.

**Quality Of The Limitations Section:**

3

**Questions For Rebuttal:**

My primary issue is with the evaluations and I would recommend the authors perform more trials in sim to obtain more reasonable variance bounds and also perform a comparative study with the real robot to strengthen their claims.

**Robotics Focus:**

3

**Summary Of Paper:**

This paper introduces a hybrid Key-state guided Online Imitation (KOI) learning approach to improve the efficiency of online imitation learning. The authors claim that this addresses the challenge of limited expert trajectories in extensive exploration spaces. The KOI method decomposes tasks into "what to do" and "how to do" aspects, utilizing visual-language models to identify semantic key states for the former and optical flow analysis to determine motion key states for the latter. This dual approach enhances task-aware exploration by refining trajectory-matching reward computations. Experiments in Meta-World and LIBERO environments show that KOI significantly boosts sample efficiency compared to previous methods, and its effectiveness is further validated in real-world robotic manipulation tasks.

**Summary Of Recommendation:**

I believe the authors' idea is interesting. However, the experiments have to be more extensive to provide clear results.

---

### Author Rebuttal · Authors · 2024-08-11

We would like to express our sincere appreciation to all reviewers for their insightful and comprehensive feedback. We sincerely appreciate the comments that the idea is interesting (Reviewer 5oNU, APBJ, 3bAV and Area Chair), the presentation is clear (Reviewer 5oNU, 3bAV, raZY and Area Chair).

To comprehensively respond to the comments of the reviewers, we provide detailed analysis of experiment results, as well as real-world videos in the supplementary materials. The table of contents is as follows, please refer to it:

- Section A: Refer to the reviewers 5oNU,APBJ and 3bAV, we provide the details and results of the real-world experiments
- Section B: Refer to reviewer 5oNU, we present the analysis of the variability and the corresponding results
- Section C: Refer to the reviewers APBJ,raZY, and 3bAV, we provide a more comprehensive ablation study
- Section D: Refer to reviewer raZY, we provide the results of the non-hierarchical task
- Section E: Refer to reviewer 3bAV, we analyze the impact of demonstration quantity

We hope our responses can address your concerns.

---

### Decision · Program_Chairs · 2024-09-04

**Decision:**

Accept

**Comment:**

The presented approach to improve online imitation learning is very interesting and generally well presented. However, some design choices could be better motivated and discussed in more detail. The reviewers highlighted several issues that should be addressed to further improve the paper. For instance, additional clarification regarding the large variance in KOI’s performance would improve the discussion of the evaluations. Furthermore, the real-world experiments are not described very well and raise some concerns regarding the claims made. The paper also lacks comparison to other hierarchical or subgoal-based methods.

## Post Rebuttal
The rebuttal addressed all major concerns and further improved the already strong work. The reviewers unanimously agree that this work should be presented at CoRL 2024.